# Ribophorin II Overexpression Is Associated with Poor Response to Induction Chemotherapy with Docetaxel, Cisplatin, and Fluorouracil in P16-Negative Locally Advanced Head and Neck Squamous Cell Carcinoma

**DOI:** 10.3390/jcm10184118

**Published:** 2021-09-12

**Authors:** Wei-Shan Chen, Sheng-Dean Luo, Tai-Jan Chiu, Yu-Ming Wang, Wei-Chih Chen, Chih-Yen Chien, Fu-Min Fang, Tai-Lin Huang, Shau-Hsuan Li

**Affiliations:** 1Department of Otolaryngology, Kaohsiung Chang Gung Memorial Hospital and Chang Gung University College of Medicine, Kaohsiung 833, Taiwan; b0002066@cgmh.org.tw (W.-S.C.); rsd0323@cgmh.org.tw (S.-D.L.); jarva@adm.cgmh.org.tw (W.-C.C.); cychien3965@cgmh.org.tw (C.-Y.C.); 2Graduate Institute of Clinical Medical Sciences, College of Medicine, Chang Gung University, Taoyuan 333, Taiwan; kuerten@cgmh.org.tw; 3Department of Hematology-Oncology, Kaohsiung Chang Gung Memorial Hospital and Chang Gung University College of Medicine, Kaohsiung 833, Taiwan; victor99@cgmh.org.tw; 4Department of Radiation Oncology, Kaohsiung Chang Gung Memorial Hospital and Chang Gung University College of Medicine, Kaohsiung 833, Taiwan; scorpion@cgmh.org.tw (Y.-M.W.); fang2569@cgmh.org.tw (F.-M.F.)

**Keywords:** head and neck cancer, ribophorin II (RPN2), induction chemotherapy

## Abstract

This study aims to evaluate the relationship between human ribophorin II (*RPN2*) and the effect of treatment using induction therapy with docetaxel, cisplatin, and fluorouracil (TPF) for *p*-16 negative locally advanced head and neck squamous cell carcinoma (HNSCC). A total of 203 patients with locally advanced p-16 negative HNSCC who received induction chemotherapy with TPF at the Kaohsiung Chang Gung Memorial Hospital between 2009 and 2014 were enrolled. Immunohistochemistry (IHC) for RPN2 was examined and correlated with treatment outcome. Our study showed that RPN2 overexpression was significantly correlated with a poor response to induction chemotherapy with TPF. Both RPN2 overexpression and clinical N1 to N3 stages represented adverse prognostic factors for progression-free survival (PFS) and overall survival (OS). RPN2 might be a predictive marker for treatment response to induction chemotherapy. Further clinical trials are needed to determine the therapeutic significance of RPN2 in patients with HNSCC.

## 1. Introduction

Head and neck squamous cell carcinoma (HNSCC) has a global incidence of more than 500,000 newly diagnosed cases annually [1]. In Taiwan, it is the fourth most common cancer and also has the fourth highest mortality rate among Taiwanese male patients, with approximately 7000 new cases diagnosed and approximately 3000 deaths annually. The median age of male patients is 56 years of age and is 62 years of age in female patients [2]. A majority of patients present with locally advanced, stage III-IV disease, with low surgical curability or inoperable status. Among all the HNSCC patients, over 70% are human papillomavirus (HPV)-negative and are generally considered to have worse survival and response to treatments such as chemotherapy than HPV-positive HNSCC patients [3], highlighting the importance of resistance to chemotherapy in HPV-negative HNSCC.

Induction chemotherapy has been one of the treatment modalities for patients with locally advanced HNSCC to achieve tumor shrinkage and to decrease the risk of distant metastasis, thereby improving survival and preserving vital organs [4]. An induction chemotherapy regimen with docetaxel, cisplatin, and fluorouracil was approved for patients with either operable or inoperable HNSCCs, based on randomized trials that revealed significantly better progression-free survival (PFS) and overall survival (OS) in patients receiving additional docetaxel than those receiving a conventional regimen comprising only cisplatin and fluorouracil [5,6]. Although induction chemotherapy with TPF showed a good response rate for HNSCC, 30–40% HNSCC patients did not respond well. Moreover, in some patients, the combination regimen of docetaxel, cisplatin, and fluorouracil increased the incidence of febrile neutropenia, stomatitis, and diarrhea [4], leading to severe intolerance, and therefore, treatment had to be aborted. Consequently, it is crucial to identify patients who may show a poor response to induction therapy in order to avoid toxic side effects and to not delay alternative therapeutic choices. Therefore, it is crucial to develop a marker that can reliably predict the response to induction therapy with TPF. 

The human ribophorin II (RPN2) gene encodes an integral rough endoplasmic reticulum (ER) glycoprotein that participates in the translocation and maintenance of rough ER structures. It has also been demonstrated that the RPN2 protein is a component of an N-oligosaccharyl transferase complex. Honma et al. [7] revealed that silencing RPN2 induces a hypersensitive response of human breast tumor cells to docetaxel. It was proven that the amount of RPN2 gene product is related to drug-resistance in tumor cells, which indicates that RPN2 might be a candidate marker to predict cancer. Studies on the relationship between RPN2 and breast cancer/gastric cancer have been published; however, until now, there have been no studies on the association between RPN2 and locally advanced or advanced head and neck squamous cell carcinoma [7,8]. 

Thus, this study aims to evaluate the relationship between RPN2 and the effect of treatment using induction therapy with TPF for locally advanced head and neck squamous cell carcinoma.

## 2. Materials and Methods

### 2.1. Patient Population

Patients with locally advanced stage III and IV p-16 negative head and neck squamous cell carcinoma who received induction chemotherapy with TPF due to technical unresectability, low surgical curability, or organ preservation at the Kaohsiung Chang Gung Memorial Hospital between 2009 and 2014 were reviewed retrospectively. Patients with distant metastasis were excluded. This study was approved by the Institutional Review Board of Chang Gung Memorial Hospital. The enrolled patients were required to provide pretreatment specimens to conduct immunohistochemical (IHC) staining. Clinical TNM staging was determined according to the 7th American Joint Committee on Cancer (AJCC) staging system. Patients eligible for induction chemotherapy with TPF met the following criteria: Eastern Cooperative Oncology Group performance status 0 or 1, age ≥ 18 years, normal bone marrow function (absolute neutrophil count ≥ 1.75 × 109/L, platelet count ≥ 100 × 109/L), hepatic function (serum total bilirubin ≤ 1.5 mg/dL, serum aspartate aminotransferase and alanine aminotransferase ≤ 2.5 × upper normal limit), and renal function (serum creatinine ≤ 1.5 mg/dL). The treatment following induction chemotherapy with TPF was chemoradiotherapy, with the exception of patients with progressive disease after induction chemotherapy. Finally, we recruited 203 patients for this retrospective study. Among these 203 patients, 197 patients received chemoradiotherapy after induction chemotherapy, and 6 patients did not receive chemoradiotherapy due to progressive disease after induction chemotherapy.

### 2.2. Treatment and Response Evaluation

Docetaxel was administered at doses of 60 to 65 mg/m^2^ by intravenous infusion for 1.5 h, cisplatin at 60 to 75 mg/m^2^ for 4 h, and 5-fluorouracil (5-FU) at 600 to 750 mg/m^2^ per 24 h with a 96-h continuous intravenous infusion. The dose-modified TPF regimen was repeated every 3 weeks for 2–3 cycles. Radiation therapy was administered at 3 to 6 weeks after the last cycle of induction chemotherapy, lasting for 7 weeks with a total dose of 66–70 Gy. It was administered concurrently with a weekly cisplatin dose of 40 mg/m^2^ via intravenous infusion. We assessed tumor responses by performing clinical status and imaging studies according to the Response Evaluation Criteria in Solid Tumors guideline version 1.0. PFS was calculated from the date of the first induction of chemotherapy to the date of progression or death. OS was calculated from the date of diagnosis until death or the last follow-up.

### 2.3. Immunohistochemical (IHC) Staining

IHC staining was performed using the immunoperoxidase technique. Staining was performed on slides (4 mm) of formalin-fixed, paraffin-embedded tissue sections using primary antibodies against RPN2 (HPA008297, 1:300, Sigma-Aldrich, St. Louis, MO, USA) and p16INK4 (BD Biosciences, San Jose, CA, USA). Briefly, after deparaffinization and rehydration, antigen retrieval was performed by treating the slides with 10 mmol/L citrate buffer (pH 6.0) in a hot water bath (95 °C) for 20 min. Endogenous peroxidase activity was blocked for 15 min with 0.3% hydrogen peroxide. After blocking with 1% goat serum for 1 h at room temperature, the sections were incubated with primary antibodies overnight for at least 18 h at 48 °C. Immunodetection was performed using the LSAB2 kit (Dako, Carpinteria, CA, USA) followed by a reaction with 3-3′-diaminobenzidine for color development, and hematoxylin was used for counterstaining. Slides incubated without primary antibodies were used as a negative control. Slides with sections of lung adenocarcinoma [9] tissue were used as a positive control. Staining assessment was independently conducted by two pathologists who did not have any information about the clinicopathological features or prognosis of the patients. A semi-quantitative immunoreactive score (IRS) was used to evaluate the RPN2 immunohistochemical staining [10]. The IRS was calculated by multiplying the staining intensity (graded as: 0 = no staining, 1 = weak staining, 2 = moderate staining, and 3 = strong staining) and the percentage of positively stained cells (0 = no stained cell, 1 ≤ 10% of stained cells, 2 = 10–50% of stained cells, 3 = 51–80% of stained cells, and 4 ≥ 80% of stained cells). The criterion for high RPN2 expression was a specimen with an IRS ≥6. Positive p16 expression was defined as strong nuclear staining in 70% or more of the tumor cells [11,12].

### 2.4. Statistical Analysis

Statistical analysis was performed using a SPSS 22 software package. The chi-square test or Fisher’s exact test was used to compare the data between two groups. Logistic models were used to evaluate the relationship between RPN2 expression and the response to induction chemotherapy with TPF. For survival analysis, the Kaplan–Meier method was used for univariate analysis, and the difference between the survival curves was tested using a log-rank test. In a stepwise forward fashion, significant parameters at the univariate level were entered into the Cox regression model to analyze their relative prognostic importance. For all of the analyses, two-sided tests of significance were used, with *p* < 0.05 considered as significant.

## 3. Results

### 3.1. Patient Characteristics

A total of 203 patients (192 men and 11 women) with a median age of 52 years (range, 29–82 years) were enrolled in this study. The patient characteristics are summarized in Table 1. Among the 203 patients with p16-negative locally advanced head and neck squamous cell carcinoma, the tumors in 16 patients (8%) were categorized as AJCC 7th stage III disease, 89 patients (44%) as stage IVA, and 98 (48%) as stage IVB. Most patients had a primary tumor site in the oral cavity (*n* = 88, 43%), followed by the oropharynx (*n* = 63, 31%). Sixty-eight patients (34%) had a T4a class tumor, and 79 patients (39%) had T4b tumor. Clinical N classification showed an N1 class in 24 patients (12%), N2 in 115 patients (56%; 5 in N2a, 58 in N2b, and 52 in N2c), and N3 in 22 patients (11%). The number of patients with a low or an over-expression of RPN2 was similar, with 100 patients (49%) showing low expression and 103 patients (51%) displaying overexpression (Figure 1). Regarding the response rate to induction chemotherapy, 12 patients (6%) exhibited complete response, and 114 patients (56%) achieved partial response; however, 53 patients (26%) had a stable tumor and 24 patients (12%) showed progression. At the time of the last analysis, the patients were followed up with for a minimum of 62 months. The median follow-up period was 90 months (range, 62–128 months) for 50 survivors, and the median PFS and OS were 16 and 26 months, respectively. Furthermore, the 5-year PFS and OS rates were 31% and 35%, respectively.

### 3.2. Relationship between RPN2 Expression, the Response of Induction Chemotherapy with TPF, and Clinicopathologic Parameters

RPN2 expression was not significantly associated with any clinical parameters, such as age, sex, primary tumor site, clinical AJCC 7th stage, clinical T and N classification, smoking history, betel nut chewing, and alcohol use history (Table 2). As for the response to induction chemotherapy, the overexpression of RPN2 was significantly associated with poor response to induction chemotherapy (*p* = 0.01). Patients with low RPN2 expression showed an overall response rate (complete response plus partial response) of 71%, while those with overexpression exhibited a worse overall response rate of 53%. The T4 tumor stage (including T4a and T4b) was also significantly related to a poor response to induction chemotherapy (*p* = 0.043) compared to the T1 to T3 stages. Other parameters, including age, sex, clinical N classification, clinical AJCC 7th stage, and primary tumor site, were not significantly correlated with the response to induction chemotherapy with TPF (Table 3). We also performed a logistic model to evaluate the relationship between RPN2 expression and the response to induction chemotherapy with TPF. RPN2 overexpression was significantly correlated with a poor response to induction chemotherapy with TPF (*p* = 0.022, OR = 2.037, 95% CI = 1.109–3.743).

### 3.3. Survival Analyses

Correlations of patient survival with various clinicopathological factors at the univariate level are shown in Table 4. The 5-year PFS and OS rates were 42% and 46% in patients with low RPN2 expression, and 20% and 24% in patients with RPN2 overexpression, respectively. RPN2 overexpression (*p* = 0.001, Figure 2A), clinical stage T4 (*p* = 0.036), clinical stages N1 to N3 (*p* = 0.021), and 7th AJCC stage IVB (*p* = 0.034) were significantly associated with inferior 5-year PFS. For 5-year OS, RPN2 overexpression (*p* = 0.002, Figure 2B), clinical stage T4 (*p* = 0.03), clinical stages N1 to N3 (*p* = 0.006), and 7th AJCC stage IVA + IVB (*p* = 0.039) were found to be significantly correlated with worse OS. In multivariate analysis, RPN2 overexpression (PFS: *p* = 0.008, OR = 1.579, 95% CI = 1.126–2.213; OS: *p* = 0.016, OR = 1.522, 95% CI = 1.081–2.141) and clinical stages N1 to N3 (PFS: *p* = 0.018, OR = 1.684, 95% CI = 1.095–2.592; OS: *p* = 0.006, OR = 1.870, 95% CI = 1.197–2.920) remained significant, and both represented adverse prognostic factors for poor PFS and OS.

Then, we performed the subgroup analysis in patients with clinical T4 disease, clinical N1~3 disease, or both clinical T4 and N1~3 disease. Among the 147 patients with clinical T4 disease, the 5-year PFS (*p* = 0.007) and OS (*p* = 0.019) rates were 38% and 43% in 68 patients with low RPN2 expression and 18% and 23% in 79 patients with RPN2 overexpression, respectively. Among the 161 patients with clinical N1~3 disease, the 5-year PFS (*p* = 0.005) and OS (*p* = 0.009) rates were 36% and 40% in the 77 patients with low RPN2 expression and 19% and 21% in the 84 patients with RPN2 overexpression, respectively. Among 115 patients with both clinical T4 and N1~3 disease, the 5-year PFS (*p* = 0.027) and OS (*p* = 0.041) rates were 31% and 37% in the 51 patients with low RPN2 expression and 16% and 19% in the 64 patients with RPN2 overexpression, respectively.

## 4. Discussion

RPN2 overexpression has been reported to have a negative effect in several human cancers. Honma et al. [7] first reported that the downregulation of RPN2 gene induces apoptosis in docetaxel-resistant breast cancer cells in the presence of docetaxel, indicating that RPN2 might be a new target for patients who show poor response to treatment with docetaxel-based neoadjuvant or adjuvant chemotherapy. Kurashige et al. [8] further demonstrated that RPN2 suppression could change the susceptibility of esophageal squamous cell carcinoma cells to docetaxel in vitro. In other in vitro studies, RPN2 was also found to exhibit exclusive co-expression with CD24 in late-stage and high-grade pancreatic adenocarcinoma cancer cells with increased invasion ability [13]. Similar studies later reported that the downregulation of RPN2 inhibits cell proliferation and further suppresses cell invasion and migration in colorectal cancer cells [14] and in non-small-cell lung cancer cells [9]. Hong et al. [15] examined nasopharyngeal carcinoma cells and found that RPN2 was overexpressed in NPC tissue; using small interfering RNA (siRNA) to silence RPN2 expression markedly decreased the migration and invasion of the NPC cells. To the best of our knowledge, the significance of RPN2 in patients with locally advanced HNSCC receiving induction chemotherapy with TPF remains unclear; therefore, we conducted the present study. In our study, a significant inverse correlation between RPN2 overexpression and the effect of treatment with TPF induction therapy was observed. After induction therapy with the TPF regimen, the overall response rate in our study was 62%. We found that patients with low RPN2 expression showed an overall response rate (CR plus PR) of up to 71%, while those with overexpression exhibited a worse overall response rate of 53%. The logistic model also showed a significant correlation between poor responses to induction chemotherapy with TPF and RPN2 overexpression (*p* = 0.022, OR = 2.037, 95% CI = 1.109–3.743). Based on Honma’s study [7], RPN2 was thought to be the cause of resistance to docetaxel, and our results further support this finding.

Previous studies on HNSCCs have indicated that an advanced N stage is related to PFS and OS. An advanced N stage is considered to be related to a higher risk of distant metastasis [16], and our results also support this view. Our univariate log-rank analysis of prognostic factors for PFS and OS revealed that for patients at the N0 stage receiving induction chemotherapy, the PFS and OS were better than those at N1–N3 stages (*p* = 0.021 for PFS, 0.006 for OS). In addition, we also found that the T4 stage, AJCC 7th stage IV, and RPN2 expression were significantly related to PFS and OS. Brockstein et al. [17] also concluded that in patients who had received induction chemotherapy, the T4 stage group exhibited worse PFS and OS than the T0–T3 groups and that the T4 stage was more likely to have locoregional recurrence than the other T stages. A few studies have assessed the relationship between RPN2 expression and clinical disease outcomes; however, they were mainly conducted in gastroenterology cancers. It was demonstrated that in patients with advanced esophageal squamous cell carcinoma [8] and in those with advanced gastric cancer [18], the RPN2-negative group had better pathological and clinical responses to docetaxel-based chemotherapy than the RPN2-positive group. RPN2-negative patients with advanced gastric cancer also showed significantly higher PFS and OS [18]. However, the relationship between RPN2 expression and treatment response to chemotherapy with TPF in patients with locally advanced head and neck squamous cell carcinoma remains unclear. In the univariate log-rank analysis in our study, the group that showed overexpression of RPN2 was related to poor outcome after TPF induction chemotherapy (*p* = 0.001 for PFS, 0.002 for OS). This is compatible with the hypothesis regarding the relationship between RPN2 and resistance to chemotherapy. For multivariate Cox regression analysis, N1–N3 classification and the overexpression of RPN2 were included in our study. The N1–N3 classification may indicate undetected distant metastasis, and it is also related to a poor response to induction chemotherapy. RPN2 expression is independent of advanced T stage, advanced N stage, and latent cancer staging and is related to prognosis following treatment with a TPF regimen. Based on our results, RPN2 gene expression may be an indicator of induction treatment outcome.

Locally advanced tumors are commonly diagnosed in patients with HNSCCs. After the TAX324 study [5,6], induction chemotherapy with TPF is frequently administered to patients with locally advanced HNSCC. After induction chemotherapy with TPF, 60–70% of patients with locally advanced HNSCC have significant tumor shrinkage [5,6]. Previous reports [19,20,21] have revealed that patients with HNSCC who responded well to induction chemotherapy had a superior prognosis compared to those who responded poorly to induction chemotherapy, indicating that induction chemotherapy may only benefit a limited proportion of HNSCC patients. On the contrary, a phase III trial by Hitt et al. [22] failed to show any advantage of induction chemotherapy with TPF over concurrent chemoradiotherapy alone in patients with unresectable locally advanced HNSCC. Ineffective induction chemotherapy may result in severe toxic effects and may cause a delay in resorting to other therapeutic options, such as concurrent chemoradiotherapy. Therefore, it is important to identify factors that can predict tumor response to define a better selection of HNSCC patients for TPF induction chemotherapy.

However, there are certain limitations of our study. First, RPN2 expression was not observed to be associated with other clinical parameters; however, these results may have been underestimated due to the relatively small sample size. A larger sample size is required for a more accurate conclusion. Second, our study was a retrospective study, and it only included single-center data. More data are needed to confirm the relationship between RPN2 and TPF regimen. Third, in the present study, patients with locally advanced HNSCC received induction chemotherapy with TPF followed by concurrent chemoradiotherapy, and we found that patients with RPN2 overexpression had poor PFS and OS. However, we only evaluated the association between RPN2 expression with induction chemotherapy response. We did not investigate the association between RPN2 expression with radiotherapy response, which also had a large impact on the treatment outcome. The poor PFS and OS in patients with RPN2 overexpression could not be attributed only to a poor response to induction chemotherapy in patients with RPN2 overexpression. Finally, our patient group showed a male predominance with 192 men and only 11 women, which may cause possible selection bias.

## 5. Conclusions

Our study showed that RPN2 overexpression was significantly correlated with a poor response to induction chemotherapy with TPF. For survival analysis, both RPN2 overexpression and clinical N1 to N3 stages significantly represented adverse prognostic factors for poor PFS and OS. Our results suggest that RPN2 might be a predictive marker for treatment response to induction chemotherapy with TPF in patients with p16-negative locally advanced head and neck squamous cell carcinoma. Further molecular studies and clinical trials are needed to determine the therapeutic significance of RPN2 in patients with HNSCC.

## Figures and Tables

**Figure 1 jcm-10-04118-f001:**
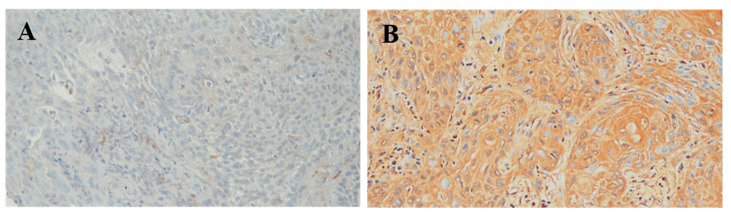
*RPN2* immunohistochemical staining of in head and neck squamous cell carcinoma. (**A**) Illustration of low *RPN2* expression in head and neck squamous cell carcinoma; original magnification ×200. (**B**) Illustration of *RPN2* overexpression in head and neck squamous cell carcinoma; original magnification ×200.

**Figure 2 jcm-10-04118-f002:**
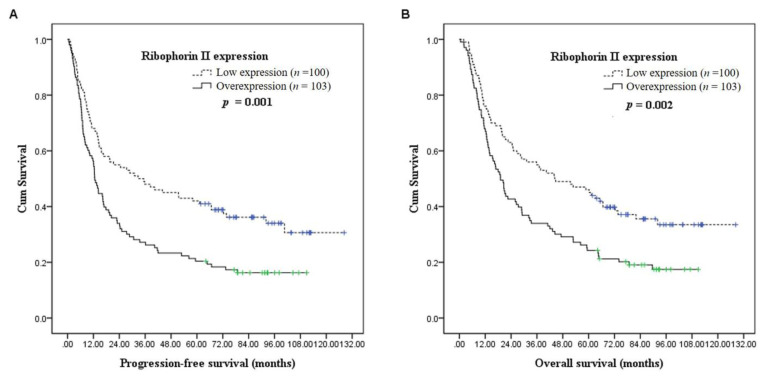
(**A**) Progression-free survival according to ribophorin II expression. (**B**) Overall survival according to ribophorin II expression.

**Table 1 jcm-10-04118-t001:** Characteristics of 203 patients with p16-negative locally advanced head and neck squamous cell carcinoma receiving induction chemotherapy with TPF ^1^.

Parameters		*n* (%)
Age		
	median	52
	mean	52
	range	29–82
Sex		
	male	192 (95%)
	female	11 (5%)
Primary tumor site		
	hypopharynx	30 (15%)
	larynx	22 (11%)
	oropharynx	63 (31%)
	oral cavity	88 (43%)
Clinical T classification		
	T1	5 (2%)
	T2	23 (11%)
	T3	28 (14%)
	T4a	68 (34%)
	T4b	79 (39%)
Clinical N classification		
	N0	42 (21%)
	N1	24 (12%)
	N2a	5 (2%)
	N2b	58 (28%)
	N2c	52 (26%)
	N3	22 (11%)
Clinical 7th AJCC stage		
	III	16 (8%)
	IVA	89 (44%)
	IVB	98 (48%)
Betel nut chewing		
	Absent	44 (22%)
	Present	159 (78%)
Smoking		
	Absent	16 (8%)
	Present	187 (92%)
Alcohol		
	Absent	33 (16%)
	Present	170 (84%)
Ribophorin II expression		
	Low expression	100 (49%)
	Overexpression	103 (51%)
Response to induction chemotherapy		
	Complete response	12 (6%)
	Partial response	114 (56%)
	Stable disease	53 (26%)
	Progression disease	24 (12%)

^1^ TPF, docetaxel, cisplatin, and fluorouracil.

**Table 2 jcm-10-04118-t002:** Associations between ribophorin II expression and clinicopathologic parameters in 203 patients with p16-negative locally advanced head and neck squamous cell carcinoma receiving induction chemotherapy with TPF ^1^.

Parameters		Ribophorin II Expression		
		Low	Over	*p* Value
Age	<52 y/o	45	55	0.23
	≥52 y/o	55	48	
Sex	Male	96	96	0.38
	Female	4	7	
Clinical T classification	T1~3	32	24	0.17
	T4	68	79	
Clinical T classification	T1~4a	63	62	0.68
	T4b	37	41	
Clinical N classification	N0	23	19	0.42
	N1~3	77	84	
Clinical N classification	N0~1	33	33	0.88
	N2~3	67	70	
Clinical 7th AJCC stage	III	10	6	0.27
	IVA, IVB	90	97	
Clinical 7th AJCC stage	III, IVA	53	52	0.72
	IVB	47	51	
Primary tumor site	Oral cavity	40	48	0.34
	Others	60	55	
Primary tumor site	Larynx/Hypopharynx	29	23	0.28
	Others	71	80	
Primary tumor site	Oropharynx	31	32	0.99
	Others	69	71	
Betel-nut chewing	Absent	23	21	0.65
	Present	77	82	
Smoking history	Absent	11	5	0.11
	Present	89	98	
Alcohol history	Absent	20	13	0.15
	Present	80	90	

^1^ TPF, docetaxel, cisplatin, and fluorouracil; x2 test, Fisher’s exact test, or t test were used for statistically analyzed.

**Table 3 jcm-10-04118-t003:** Associations between the response to induction chemotherapy with TPF ^1^ and clinicopathologic parameters in 203 patients with p16-negative locally advanced head and neck squamous cell carcinoma.

Parameters		Response to Induction Chemotherapy		
		CR/PR ^2^	SD/PD ^3^	*p* Value
Age	<52 y/o	64	36	0.58
	≥52 y/o	62	41	
Sex	Male	119	73	1.00
	Female	7	4	
Ribophorin II expression	Low	71	29	0.01 *
	Over	55	48	
Clinical T classification	T1~3	41	15	0.043 *
	T4	85	62	
Clinical T classification	T1~4a	80	45	0.47
	T4b	46	32	
Clinical N classification	N0	30	12	0.16
	N1~3	96	65	
Clinical N classification	N0~1	46	20	0.12
	N2~3	80	57	
Clinical 7th AJCC stage	III	13	3	0.099
	IVA, IVB	113	74	
Clinical 7th AJCC stage	III, IVA	69	36	0.27
	IVB	57	41	
Primary tumor site	Oral cavity	48	40	0.053
	Others	78	37	
Primary tumor site	Larynx/Hypopharynx	34	18	0.57
	Others	92	59	
Primary tumor site	Oropharynx	44	19	0.13
	Others	82	58	
Betel-nut chewing	Absent	30	14	0.35
	Present	96	63	
Smoking	Absent	11	5	0.57
	Present	115	72	
Alcohol	Absent	22	11	0.55
	Present	104	66	

^1^ TPF, docetaxel, cisplatin, and fluorouracil; ^2^ CR, complete response, PR, partial response; ^3^ SD, stable disease, PD, progressive disease. * Statistically significant. x2 test, Fisher’s exact test, or t test were used for statistically analyzed.

**Table 4 jcm-10-04118-t004:** Results of univariate log-rank analysis of prognostic factors for progression-free survival and overall survival in 203 patients with p16-negative locally advanced head and neck squamous cell carcinoma receiving induction chemotherapy with TPF ^1^.

Factors	No. of Patients	Progression-Free Survival (PFS)		Overall Survival (OS)	
		5-Year PFS Rate (%)	*p* Value	5-Year OS Rate (%)	*p* Value
Age					
<52 y/o	100	25%	0.14	30%	0.20
≥52 y/o	103	37%		40%	
Sex					
Male	192	31%	0.99	35%	0.89
Female	11	27%		36%	
Ribophorin II expression					
Low expression	100	42%	0.001 *	46%	0.002 *
Overexpression	103	20%		24%	
Clinical T classification					
T1~3	56	41%	0.036 *	43%	0.03 *
T4	147	27%		32%	
Clinical T classification					
T1~4a	125	34%	0.12	37%	0.14
T4b	78	27%		32%	
Clinical N classification					
N0	42	45%	0.021 *	52%	0.006 *
N1~3	161	27%		30%	
Clinical N classification					
N0~1	66	42%	0.012 *	50%	0.005 *
N2~3	137	26%		28%	
Clinical 7th AJCC stage					
III	16	50%	0.083	56%	0.039 *
IVA, IVB	187	29%		33%	
Clinical 7th AJCC stage					
III, IVA	105	35%	0.034 *	38%	0.043 *
IVB	98	27%		32%	
Primary tumor site					
Oral cavity	88	30%	0.24	33%	0.27
Others	115	32%		37%	
Primary tumor site					
Larynx/Hypopharynx	52	40%	0.085	46%	0.082
Others	151	28%		31%	
Primary tumor site					
Oropharynx	63	25%	0.64	29%	0.58
Others	140	34%		38%	
Betel-nut chewing					
Absent	44	39%	0.21	43%	0.31
Present	159	29%		33%	
Smoking					
Absent	16	38%	0.33	44%	0.31
Present	187	31%		34%	
Alcohol					
Absent	33	46%	0.079	49%	0.093
Present	170	28%		32%	

^1^ TPF, docetaxel, cisplatin, and fluorouracil; * Statistically significant.

## Data Availability

The data presented in this study are available on request from the corresponding author.

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
