# Peer review of "Ribophorin II Overexpression Is Associated with Poor Response to Induction Chemotherapy with Docetaxel, Cisplatin, and Fluorouracil in P16-Negative Locally Advanced Head and Neck Squamous Cell Carcinoma"

_jcm, 2021, doi:10.3390/jcm10184118_

Round 1
Reviewer 1 Report
First of all, I would like to thank the editorial office for the opportunity to evaluate this interesting work.
The authors conducted this study, whose objective was to evaluate the relationship between RPN2 and the effect of induction therapy treatment for locally advanced squamous cell carcinoma of the head and neck, in an accurate and methodologically correct manner.
The introduction is relevant and gives an appropriate bibliography.
Materials and methods are described clearly and completely.
The results are accurately reported. Statistical analysis is appropriate.
The discussion is relevant. However, in lines 219-220 is cited an article (4) comparing two groups of patients with advanced carcinoma of the head and neck, treated by two different patterns of induction chemotherapy. The results of the cited work could not be compared with the results of this manuscript.
The interesting conclusions are supported by the results.
the following note have to add the previous review of the manuscript "Ribophorin II overexpression is associated with poor response to
induction chemotherapy with docetaxel, cisplatin, and fluorouracil in p16-negative locally advanced head and neck squamous cell carcinoma".
The title refers to the association between Ribophorin expression and induction chemotherapy response while the study considers the progression free survival and Overall survival after induction chemotherapy plus radiotherapy. Radiotherapy influence the PFS and OS, for example we don’t know if patients no responder to induction chemotherapy had a good response to radiotherapy and then a good PFS and OS. Then the correlation between RPN2 overexpression and poor PFS and OS could not to be attribute only to the poor response to the induction chemotherapy, because we don’t know what was the response to the radiotherapy.
Furthermore, the authors reports a poor PFS and OS correlated with T4, N+ and RPN2 overexpression , to better understand the role of RPN2 on the PFS and OS in T4 and N+ patients should interesting to show the PFS and OS in T4 and in N+ tumors with RPN2 overexpression and low expression.
Finally , the author should explain why they included in the study T1 and T2 patients.
Author Response
Dear reviewers:
Please see the attachment, thanks!

Reviewer 2 Report
The aim of the submitted paper is interesting, as the prospected role of Ribophorin II can be considered a future important prognostic factor for induction chemotherapy in patients affected by HNSCC. However, some issues are herein suggested.
1) In the introduction subheading more detailed data should be reported about the incidence of HNSCC, especially mentioning the mean/median ages of people affected by this cancer.
2) All patients were affected by locally advanced HNSCC stage III and IV. Were there patients with distant metastasis in the study population? Please specify.
3) All patients undergo RT-CH 3-6 weeks after the last cycle of induction chemoteraphy. On what basis these patients were candidated for chemoradiation tretment? Were they non-operable HNSCCs? Please specify.
4) The submitted study reports that 192 men and only 11 women were enrolled. This gender difference should be considered as a potential selection bias in the discussion subheading.
5) In lines 181-182 it is reported that “in patients with RPN2 overexpression, the 5-year PFS and OS were 20% and 24%, respectively”. It is recommended that also data about PFS and OS in patients with RPN2 low expression (42% PFS - 46% OS) are added in the text (at present such data are available in a separate table only) in order to better underline such important results.
6) Line 66: …however until now, there are no studies on the association between RPN2 and locally advanced or advanced head and neck….
7) Line 69: …and the effect of treatment using induction therapy with TPF for locally advanced…
8) Line 211: …expression markedly decreased migration and invasion….
9) Line 271: …our study was a retrospective study and it only included single-center data…
Author Response

(The authors gave the same response as above.)
